# Epidemiological Studies of Brown Rot in Spanish Cherry Orchards in the Jerte Valley

**DOI:** 10.3390/jof7030203

**Published:** 2021-03-10

**Authors:** Inmaculada Larena, Maria Villarino, Paloma Melgarejo, Antonieta De Cal

**Affiliations:** 1Department of Plant Protection, Instituto Nacional de Investigación y Tecnología Agraria y Alimentaria, Ctra. de La Coruña Km. 7, 28040 Madrid, Spain; ilarena@inia.es (I.L.); villarino.maria@inia.es (M.V.); 2Direccion General de Producciones y Mercados Agrarios, MAPA, 28014 Madrid, Spain; pmelgarejo@mapa.es

**Keywords:** *Monilinia laxa*, prediction model, primary inoculum, relative humidity, rainfall

## Abstract

Cherry brown rot caused by *Monilinia*
*laxa* was observed and estimated in organic cherry orchard located in the Jerte Valley between 2013 and 2018 (Cáceres, Spain). Climatic variables were collected from this orchard and also from a nearby weather station. The primary inoculum of the pathogen recorded in March was detected in overwintered mummified fruits, ground mummies, and necrotic twigs and was a function of the average temperature of the previous three months (December, January, and February). The first symptoms of brown rot could be observed on flowers until fruit set in April. The months of March and April were identified as the critical period for cherry brown-rot development. A significant positive correlation was identified between brown rot observed at harvest and the mean number of consecutive days in each fortnight of March and April when the percent relative humidity was above 80%. Brown-rot incidence observed over the 6 years ranged from 0 to 38%. More than 11 days with relative humidity >80% in each fortnight of critical period would mean 100% of cherry brown rot at harvest. A forecasting model could be used to predict brown rot infection in Jerte Valley cherries.

## 1. Introduction

Cherry brown rot is caused by *Monilinia laxa* (Aderh and Ruhland) Honey and *M. fructigena* (Pers.) Honey in Jerte Valley in Spain [1], where two infection phases, blossom-blight and fruit-rot phases, can be distinguished [2,3]. The Jerte Valley is one of the largest sweet cherry (*Prunus avium* L.) growing areas in Spain with up to 7490 ha. at an altitude of between 400 and 1200 m and more than 36,000 mt of annual cherry production [4]. In the Jerte Valley, cherry cultivation has traditionally been rain fed with occasional irrigation support, and blooming and fruit ripening varies by approximately 10 to 14 days, depending on the altitude.

Symptoms of cherry blossom blight are first observed on the anthers of flowers in spring and then on the reproductive structures of the flower. Under favorable conditions, not only flowers and twigs but also fruit can produce blighted symptoms similar to those of blossom and twig blight [5]. The fungus frequently spreads into shoots, twigs, and small branches from where cankers and the mass production of gums and abundant sporulation may originate [2]. Fruit blight can occur in two ways: as a result of blossom and/or twig blight near the fruit, or at certain stages of green fruit in the absence of blossom or twig blight [5]. Fruit blight occurs between the periods of blossom blight and harvest fruit rot and often appears on the same shoots that have previously shown blossom and/or shoot blight, but also occurs separately from these symptoms [5]. Fruit-blight incidence was related to blossom blight in spring and fruit rot at harvest. The fruit can be infected by the pathogen at any stage of its development, but the disease only becomes more severe when the fruit begins to ripen [3]. The airborne density of *Monilinia* conidia increases continuously from the first appearance of infected fruit until their harvest [6,7]. Brown rot losses in cherries can be up to 33% at harvest, and after cold storage at 0 °C for one month, losses of up to 86% of rotted fruit have been reported [3].

Brown rot is spread by the dispersal of *Monilinia* conidia, which can be caused by wind, water, insects, birds, and man [2]. *Monilinia* airborne conidia are deposited on the fruit surface, where they can cause infection [8,9]. Survival, colonization, latency, reproduction, release, transport, and deposition of *Monilinia* conidia are related to the environmental temperature (T), the relative humidity (RH), the amount of rainfall (R), and the wind direction [6,10,11]. Since brown rot is a polycyclic disease, secondary inoculum is of great importance on its incidence and severity in each growing season [2]. In addition, secondary inoculum can occur anywhere in the infected tissue where the moisture content is sufficient for the pathogen to sporulate [12]. Since brown-rot outbreaks in stone fruit depend on the prevailing environmental conditions, a high RH and a T range between 15 and 25 °C [13] favor disease development, although infection can also occur under more extreme conditions (5–30 °C) [14]. The incidence of cherry-blossom infection by *M. fructicola* [15] and *M. laxa* [16] was also shown to rise with increasing wetness duration (W). In addition, a dry environment favors conidial dispersion [7], while high T and low RH reduce conidial viability [17]. This paradox suggests that conidial viability could be long lasting in the field and/or sufficient moisture exists at the infection site [7]. Positive relationships between T, wind speed, and the density of airborne conidia have also been reported [10].

When climatic conditions are unfavorable, infections can remain latent until conditions are favorable for the expression of the disease; at which point, fruit rot occurs [2]. A positive correlation has been reported between the incidence of latent infection and harvest or postharvest brown rot in cherry [18]. Using this model, Gell et al. [19] demonstrated that T and W were the two most important weather factors contributing to the incidence of latent infections caused by *M. laxa* and *M. fructigena* in Spanish peach orchards. In fact, T and W can account for more than 90% of the incidence of brown rot, and that of the two climatic factors, W is more influential than T [19]. In order for latent infections to develop, Gell et al. [19] also found that W needs to be longer than 22 h when T is 8 °C or when W is 5 h and T is 25 °C. Luo and Michailides [20] also reported that W was important for the development of latent infections: the incidence of latent infection increased linearly with increasing W during the blooming period and exponentially with increasing W during the early and late fruit development phases.

Losses in cherry production due to *Monilinia* spp. attack increase with the susceptibility of the cultivar and if the climatology is favorable to the development of brown rot. Preventive chemical treatments to reduce *Monilinia* were applied in the Jerte Valley every 7 days in rainy periods or every 10 days in dry periods [21]. However, brown rot is not controlled by phytosanitary treatments in years with very favorable weather conditions or in very susceptible cultivars. The epidemiology of brown rot may help avoid fungicidal treatments when they are not necessary.

Our main objective was to identify the relationship between cherry brown rot and weather conditions in the Jerte Valley. Thus, brown rot was related to weather conditions in a cherry orchard located at 500 m altitude in Navaconcejo (the Jerte Valley, Cáceres, Spain) for 6 years.

## 2. Materials and Methods

### 2.1. Orchard and Experimental Design

Six consecutive samplings were carried out in an organic cherry orchard located in Navaconcejo (40°9′31.06″ N–5°49′49.72″ W) (Cáceres, Spain) between 2013 and 2018 (Figure 1). The orchard of mixed late varieties (cvs. “Lapin”, “Ambrunes”, and “228”) where natural vegetation covered the soil at 500 m altitude, consisted a minimum of 150 trees, with a planting distance 4 × 4 m. An experimental plot with 10 trees was selected randomly for recording disease development in relation to the weather in each survey. Fruit from the mixed late cultivars were harvested between June and July. Standard organic commercial practices were followed, with no application of systemic fungicides.

### 2.2. Determination of Pathogen Primary Inoculum Sources

The cherry orchard was sampled the first week of March each year to determine overwintered sources of *Monilinia* spp. (Table 1). Each year, the numbers of overwintered mummified and nonabscised aborted fruit on the trees, cankers, and necrotic twigs of the 10 cherry trees in the observation area were counted. Ten randomly selected samples were collected from each potential source of primary inoculum in each sampling and taken to the laboratory for further study (see below).

In the laboratory, mummified fruit from the trees and ground, nonabscised aborted fruit, and cankers were first transferred to humidity chambers to determine the presence or absence of the pathogen. For this purpose, samples were incubated at 20 to 25 °C for 2 to 5 days. At the end of the incubation, the number of samples on which *Monilinia* conidiophores and conidia were visible was counted under stereomicroscope (×40) (Wild M3 Stereomicroscope, Gais, Switzerland). When *Monilinia* conidiophores and hyaline conidia with a limoniform shape and an approximate length of 12 μm were observed, the fungal structures were excised from samples, and the fungus was isolated on potato dextrose agar (PDA; Difco, Detroit, MI, USA) that was supplemented with streptomycin (0.5 g L^−1^). After a 5 day incubation in the dark at 20 to 25 °C, the identification of all recovered isolates of *Monilinia* spp. was based on the morphologic characteristics of the cultured fungus [22]. Atypical *Monilinia* isolates were identified by PCR with three sets of primers, ILaxaS/ILaxaAS, IColaS/IColaAS, and IGenaS/IGenaAS [23].

Each necrotic twig collected was cut into small pieces less than 5 cm in length. Each cut sample was superficially disinfected by immersing it in a 1% NaOCl solution for 5 min, and then in 70% ethanol for 1 min [24]. Each sample was then rinsed twice in sterile distilled water (SDW) before being placed on PDA supplemented with streptomycin (0.5 g L^−1^) and incubated at 20 to 25 °C in the dark for 7 days. At the end of the incubation period, the number of tissues in which *Monilinia* conidiophores and conidia developed was counted. Identification of all recovered *Monilinia* spp. isolates was performed by morphology and PCR, as described above.

### 2.3. Brown-Rot Evaluation

Brown-rot incidence was estimated by a visual count of signs of the disease on blossoms and fruit from the 10 randomly selected trees within the experimental plot. The evaluation was carried out during four growth stages: blossom (late March to early April), pink fruit color (late April to early May), harvest (June–July), and after harvest (October).

### 2.4. Monitoring of Environmental Variables

Hourly readings of T (°C), RH (%), and dew point (°C) were recorded by an automated data logger (EL-USB-2, Lascar electronic, UK) placed in the orchard at a height of 1.5 m above the ground. The mobile weather station was installed from 1 January 2013, to 31 December 2018 (Appendix A). In addition, data on T (°C), RH (%), rainfall (R) (mm), and wind speed (m/s) from the stationary weather station in Valdastilla (40°8′28.37″ N–5°52´7.73″ W, (Cáceres, Spain)) at 515 m altitude and 3.80 km from the experimental orchard in Navaconcejo were collected from 2013 to 2020 (Appendix A). Using these data, the average T, RH, and R from the first swelling time to the setting time were calculated using Microsoft Excel. The months of March and April were divided into fortnights, for which the mean values of T, RH, R, and number of days in which the RH was higher than 80% were calculated. These four fortnights of March and April were considered the critical period for *Monilinia* infection of cherries in the Jerte Valley.

### 2.5. Data Analysis

The incidence of blossom blight and brown rot for each year was analyzed by analysis of variance. The level of statistical significance was set at 5%. When the F test was significant at *p* < 0.05, the means were compared by the Student–Newman–Keuls multiple range test [25].

Correlation and regression analyses were performed with the STATGRAPHICS program (XVII Centurion. v. 17.2.00) on the combined data from all samplings to analyze the relationships between the incidence of brown rot and climatic conditions from February to May. Each point in the analysis was the mean of a study parameter in each year. The incidence of brown rot was the mean of 10 replicates. Data on the incidence of brown rot were arcsine transformed before analysis. In addition, correlation analyses were conducted to explore the relationship between brown rot incidence and climatic conditions.

## 3. Results

### 3.1. Pathogen Primary Inoculum Sources

Primary inoculum was recorded in March of each year (Figure 2a). The highest primary inoculum was recovered in 2013, followed by 2016 (Figure 2a). Sporulation and/or mycelial growth of *Monilinia* spp. were detected only on overwintered mummified fruit and necrotic twigs collected from cherry trees in each growing season, except in 2016, where the primary inoculum of the pathogen was also detected on ground mummies (Figure 2b). The highest number of mummified fruits was recorded on cherry fruit in 2013 and 2015. *Monilinia* spp. was not detected on necrotic twigs in 2014 and 2015 (Figure 2b). The highest number of infected necrotic twigs was detected in 2013 and 2016 (Figure 2b). *Monilinia* spp. was not detected in nonabscised aborted fruit or in trees cankers or pruned branches collected from orchard ground.

The only *Monilinia* species isolated in each primary inoculum during the 6 experimental years was *M. laxa*. Overwintered *M. laxa* only produced spore inoculum from mycelia without finding apothecia in any cherry trees during the experimental period.

### 3.2. Climatic Condition Effects on Primary Inoculum

T is the most important climatic parameter determining the development of the primary inoculum (r = 0.98; *p* = 0.003) (Table 1). For the germination of *M. laxa* conidia from primary inoculum, a significant positive correlation was observed between primary inoculum recorded in March and the mean T of the previous three months (December, January and February). The largest viable primary inoculum was recorded after the warm winter months. No effect on the primary inoculum due to RH or dew point was observed in the Navaconcejo conditions (Table 1).

In addition, the percentage of viable primary inoculum recorded in March was a function of the mean T of the previous three months (December, January, and February) and could be adjusted (Table 1) by the following equation:Viable primary inoculum (%) = −21.04 + 3.37 × T (°C) (R^2^ = 0.97; *p* = 0.003)(1)

This model was also fitted to T values from December to February in Valdastillas (Cáceres, Spain) from 2012 to 2020. The estimated primary inoculum percentage was always up to 5%, and only in the years 2016 and 2020 was higher than 10% (Table 2).

### 3.3. Cherry Brown Rot Development

The first symptoms of brown rot could be observed in late March or early April on the blossoms and remained there until the petals fall and fruit set in April. The first symptoms of the disease on the fruit could be observed in mid-May.

The incidence of brown rot in Navaconcejo cherry trees varied throughout the study years, peaking in 2018 (Figure 3). However, the incidence of *Monilinia* spp. maintained throughout the crop cycle in each year from blossom to harvest. During 2018, the highest incidences were recorded for both blossom blight and fruit rot. Brown-rot incidence was minimal during 2015 and 2016 (Figure 3). Only *M. laxa* was isolated from brown-rot samples throughout the 6 experimental years.

### 3.4. Effects of Climatic Conditions on Cherry Brown Rot

The RH during the months of March and April and the number of consecutive days in which the RH percentage was above 80% were the most important parameters that determined the cherry brown rot observed at harvest in Navaconcejo (Table 3). A significant positive correlation was identified between brown rot observed at harvest and the mean number of consecutive days (MCD) in each fortnight of March and April in which the percentage of RH was above 80% (MCD-RH ≥ 80) (r = 0.86; *p* = 0.02) (Table 3).

In addition, the brown rot observed at harvest (%) in Navaconcejo was a function of the mean number of consecutive days in each fortnight of March and April in which the percentage of RH was above 80% (MCD-RH ≥ 80) and could be fitted by the following equations:Brown rot at harvest (%) = −3.09 + 0.80 × (MCD-RH ≥ 80)2 (R^2^ = 73.95%; *p* = 0.02)(2)

A significant positive correlation was also identified between brown rot observed at harvest, brown rot estimated by model (2) (*r* = 0.87; *p* = 0.02) at harvest, and annual brown rot observed (*r* = 0.99; *p* = 0.0001) (Table 3).

One hundred percent of cherry brown rot at harvest was estimated by the model (2) for more than 11 days with RH >80% in each fortnight of March and April (critical period) in Navaconcejo and more than 50% of cherry brown rot for more than 7 days. More than 3 days with RH >80% in each fortnight of the critical period was required for brown rot to be present on the fruit at harvest (Table 4).

The incidence of brown rot in Valdastillas was predicted for 2013, 2014, 2016, 2017, 2018, and 2020 by model (2) using MCD-RH ≥ 80 (Table 5).

A significant positive correlation was also identified between brown rot (%) estimated at harvest and MCD-RH ≥ 80% (*r* = 0.95; *p* = 0.003), mean %RH (r = 0.91; *p* = 0.01), and mean number of days in each March and April fortnight with rain fall ≥10 mm (MD-R ≥10) (*r* = 0.86; *p* = 0.037) (Table 5).

## 4. Discussion

Cherry brown rot is a common disease caused by *M. laxa* in the Jerte Valley (Cáceres, Spain) that could be predicted by the number of consecutive days with RH above 80% during March and April. Cherries are exposed to rot, both in the orchard and in storage, by several fungal species, of which the most important in Europe are the brown-rot fungi *M. laxa* and *M. fructigena* [17,26,27]. Both species of *Monilinia* overwinter on mummified fruit, but *M. laxa* also causes cherry blossom blight, meaning that in seasons favorable to blossom blight, there is a source of inoculum for fruit rot [2,28,29,30]. Losses of up to 33% or 86% rotted fruit primarily caused by *M. laxa* and *M. fructigena* can be registered at harvest or after cold storage at 0 °C for 1 month, respectively [17]. However, the severity of brown rot in Jerte cherries was only caused by *M. laxa*, whose disease was not usually important, except in those years when more than three consecutive days of RH above 80% were recorded in each fortnight of March and April.

Weather factors play a key role in the brown-rot infection process [31]. Temperature and wetness duration are important environmental factors affecting the development of cherry blossom blight caused by *Monilinia* spp. [15]. The viability of primary inoculum and, therefore, the onset of spore dispersal were significantly influenced by the temperature in the Jerte Valley, which was the most important climatic parameter in the development of the primary inoculum. Interestingly, the results of the correlation analyses showed the relationship between the current year’s primary inocula, recorded in March, and the previous year’s December, January, and February temperatures, providing statistical support for the long-standing notion of disease progression from one year to the next [32]. In the case of spores that are mainly airborne, such as those of *Monilinia* spp. [6], the presence of a pronounced dispersal gradient would easily explain the observed association between the symptoms of the current year and those of the previous year. Transport of spores from twig inoculum sources to other susceptible canopy elements by splash [33] and rainwater runoff would also favor short distance association between current and previous year symptoms. Stensvand et al. [29] previously documented that *M. laxa* sporulation on twig cankers of sweet cherry peaks at blossom time, further supporting the role of twig cankers as a key inoculum source for inciting blossom blight. *M. laxa* conidia emitted germ tubes and/or produce appressoria at different T and RH [34]. However, its appressoria and conidia reached 60–80% germination after 48 h of incubation at 10 °C and 100% RH on fruit skin extract [34]. With the results of this study, the period of risk of brown rot was considered to begin when an average daily temperature above 10 °C was reached. In the Jerte Valley, this daily temperature was obtained from mid-February and continued until harvest. Tamm et al. [16] also showed that the most infections of sweet cherry blossoms caused by *M. laxa* occurred within the first 8 h of wetting with an inoculum concentration of 5000 conidia per mL and temperature ≥10 °C. For *M. laxa*, the maximum conidial germination occurs at 25 °C exposed at 100% RH [34,35], and up to 70% is reported at 10 °C [34]. Little or no germ tube formation by *M. laxa* conidia is recorded at 4 °C and 60% RH, or at 35 °C [34].

Cherry brown rot was correlated to RH and R during March and April in the Jerte Valley. Cherry blossom blight incidence was directly proportional to RH during the incubation period, with maximal levels developing at constant incubation RH levels >90% and relatively low incidences developing at RH levels <60% [36]. Furthermore, when RH during incubation ranged between similar extremes, the incidence of blight was proportional to the number of hours at values close to 90% [36]. A wetness duration requirement as short as 3 h may be sufficient for *Monilinia* germination and infection in cherry orchards [3]. *M. laxa* conidia can complete the germination process in 2 h and the germination percentage can reach >80% after 4 h at the optimum T (20–25°C) [37]. High rainfall may have resulted in increased inoculum production and infection of cherry blossoms and fruit [3]. A significant positive correlation was observed between the mean number of consecutive days in each fortnight of March and April when the RH percentage was above 80% (r = 0.95; *p* = 0.003), MD-R ≥10 (*r* = 0.86; *p* = 0.037), and cherry brown rot at harvest. The incidence of sour cherry blossom blight is also influenced by phenological stage of the tree [16] and inoculum concentration [15]. In the Jerte Valley, spore ejection could be recorded during a critical period when a great rainfall occurred, and conidia ejections decreased after May 15 because rain was scarcely. The peak incidence of blossom blight was 9 days after petal fall, while that of the final fruit blight incidence was 31 days after petal fall [5]. Thus, cherry fruit were infected by the spore ejections occurring at the end of April, and middle and late varieties showed the heaviest infection. Cherry varieties differ in susceptibility to rotting [38]. Cherry fruit state has also a significant effect on its susceptibility to *M. laxa* [3], where older fruit are more susceptible. Fruit are initially resistant to infection by *M. laxa* conidia until they reached the stage when fruit began coloring [3]. The relationship between the phenological stage of the fruit and susceptibility to brown rot in peaches has also been observed [39,40].

Disease forecasting has become an established component of quantitative epidemiology. Although it is difficult to predict disease incidence to an exact value, estimating a possible range of disease intensity (risk) can be relatively easy. This improvement will provide valuable information to decision makers. Cherry brown rot at harvest (%) in the Jerte Valley was a function of mean number of consecutive days in which the RH percentage was above 80% in each fortnight of March and April. The linear statistical model of cherry brown rot is a regression equation that describes the relationship between the mean number of consecutive days in each fortnight of March and April when the RH percentage was above 80% and brown rot. For example, for more than 11 consecutive days with RH >80% in each fortnight of March and April, it is estimated that 100% of the cherry crop is rotten, and at least 3 consecutive days with RH >80% in each fortnight of the critical period for the risk of rot to appear at harvest. The mathematical description of the model will prove useful in research, but this is also an easy parameter to measure in any orchard by the farmer himself, meaning it would not be necessary to consult nearby climate stations. The present model was used to predict the incidence of cherry brown rot in orchards located in the same area, where the mean number of days in each fortnight of March and April with R ≥10 mm also showed a significant correlation with cherry brown-rot incidence. Luo and Michailides [20,41] suggested a disease forecast model to predict the development of brown rot of French prune caused by *M. fructicola* by considering temperature and RH.

Overall, this work indicated that the forecasting model could be used to predict brown-rot infection of cherries in the Jerte Valley. The availability of fungicides with good curative activity against cherry brown rot may ultimately facilitate a strategy to respond to a specific set of environmental conditions after they occur, rather than relying prophylactically on weather conditions. Further research should be conducted to validate this model in other cherry areas and correlate the level of risk with brown-rot incidence to determine when fungicide application is economically justified.

## 5. Conclusions

The model discussed in this work performs as intended, simulating the potential brown-rot epidemic in the Jerte Valley. These results also suggest that cherry brown rot should be influenced not only by inoculum availability, temperature, and wet period durations, but also by humidity levels after March and April. This may have many applications, such as understanding pathosystem behavior in this area of cherry trees or knowing when fungicides may ultimately facilitate a strategy against brown rot.

## Figures and Tables

**Figure 1 jof-07-00203-f001:**
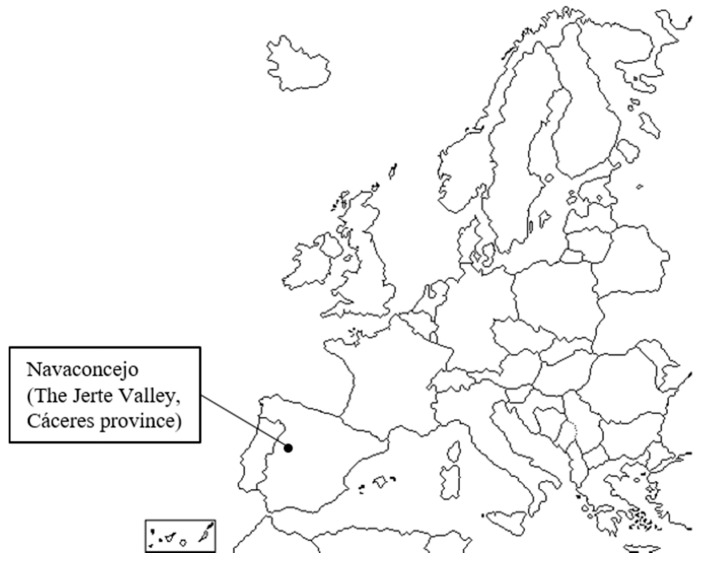
Location map of cherry orchard in Navaconcejo (Cáceres, Spain).

**Figure 2 jof-07-00203-f002:**
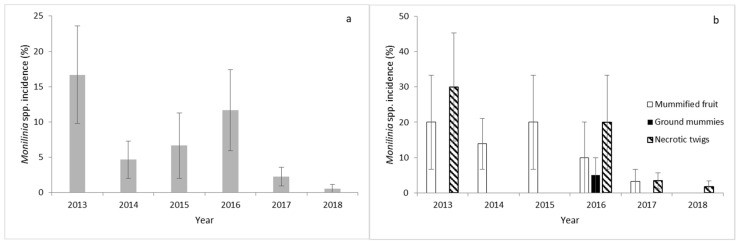
(**a**) Percentage incidence of *Monilinia* spp. in the total primary inoculum recovered in the cherry orchard of Navaconcejo (Cáceres, Spain) during 6 consecutive years. (**b**) Percentage incidence of *Monilinia* spp. in each type of primary inoculum sample (mummified fruit, ground mummies, and necrotic twigs). Data were the mean of 10 randomly selected samples from each potential source in each of the 10 cherry trees ± standard error.

**Figure 3 jof-07-00203-f003:**
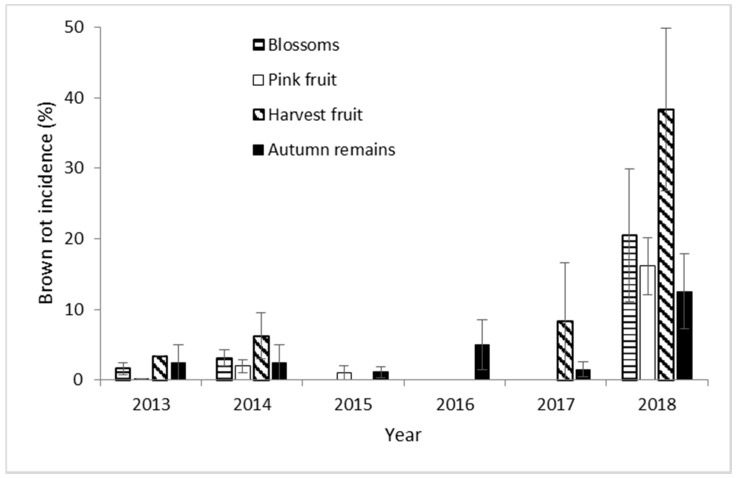
Percentage of brown-rot incidence on blossoms, pink fruit, harvest fruit, and autumn remains from cherry orchard in Navaconcejo (Cáceres, Spain) for 6 consecutive years. Data were the mean of 10 cherry trees in each year ± standard error.

**Table 1 jof-07-00203-t001:** Observed and estimated viable primary inoculum, mean temperature (T), mean relative humidity (RH), and mean dew point (°C) from December to February in the experimental cherry orchard in Navaconcejo (Cáceres, Spain).

Period	Mean T (°C)	Mean RH (%)	Dew Point (°C)	Primary Inoculum Observed (%)	Primary Inoculum (%) Estimated by Model (1)
2013–14	7.52	76.02	2.93	4.65	4.32
2014–15	8.60	59.35	0.12	6.67	7.95
2015–16	9.44	74.81	4.88	11.6	10.79
2016–17	6.91	78.65	3.22	2.25	2.26
2017–18	6.36	74.84	2.02	0.56	0.41
r ^1^	0.98	−0.27	0.39		
*p*-value ^1^	0.003	0.660	0.519		

^1^ Correlation analysis: r = correlation coefficient and *p*-value.

**Table 2 jof-07-00203-t002:** Primary inoculum estimated by model (1) and mean temperature (T) from December to February in Valdastillas (Cáceres, Spain) from 2012 to 2020.

Period	Mean T (°C)	Primary Inoculum (%) Estimated by Model (1)
2012–2013	7.75	5.09
2013–2014	8.00	5.95
2014–2015	7.83	5.37
2015–2016	9.28	10.24
2016–2017	8.89	8.94
2017–2018	7.80	5.26
2018–2019	8.58	7.88
2019–2020	9.34	10.47

**Table 3 jof-07-00203-t003:** Relationship between brown rot (BR) observed at harvest, BR estimated by model (2), annual BR, and weather conditions in each fortnight of March and April in the experimental cherry orchard of Navaconcejo (Cáceres, Spain) ^1^.

Year	Mean T (°C)	Mean RH (%)	MCD-RH ≥ 80	Annual BR Observed (%)	BR Observed at Harvest (%)	BR Estimated at Harvest (%) by Model (2)
2013	11.58	70.96	5.00	1.70	3.39	16.93
2014	13.36	63.44	3.75	3.79	6.25	8.17
2015	13.05	57.60	1.50	0.50	0.00	0.00
2016	10.12	65.09	1.50	0.00	0.00	0.00
2017	12.86	58.59	2.75	2.77	8.30	2.97
2018	9.47	73.89	6.50	24.97	38.30	30.74
r ^2^	−0.57	0.63	0.86	0.99		0.87
*p*-value ^2^	0.23	0.17	0.02	0.0001		0.02

^1^ Climatic values during critical period (March and April): temperature (T), relative humidity (RH), and mean number of consecutive days (MCD) in each fortnight of March and April in which the percentage of RH was above 80% (MCD-RH ≥ 80). ^2^ Correlation analysis: r = correlation coefficient and *p*-value.

**Table 4 jof-07-00203-t004:** Relationship between the mean number of consecutive days in each fortnight of March and April in which the RH percentage was above 80% (MCD-RH ≥ 80) and cherry brown rot (BR) estimated by model (2) at harvest in Navaconcejo (Cáceres, Spain).

MCD-RH > 80% in Each Fortnight	BR (%) Estimated by Model (2)
0	0.0
1	0.0
2	0.12
3	4.12
4	9.72
5	16.93
6	25.74
7	36.14
8	48.15
9	61.76
10	76.98
11	93.79
12	100.0
13	100.0
14	100.0
15	100.0

**Table 5 jof-07-00203-t005:** Relationship between climatic conditions in Valdastillas (Cáceres, Spain) during the critical period (March and April) and brown rot (BR) estimated by model (2) at harvest for 6 years ^1^.

Year	Mean T (°C)	Mean RH (%)	MCD-RH ≥ 80	R (mm)	MD-R ≥10	Wind Speed (m/s)	BR (%) Estimated by Model (2)
2013	11.25	67.07	5	8.00	3.75	1.74	16.91
2014	13.71	59.59	2.5	3.22	1.75	1.62	1.91
2016	10.81	63.61	3.5	4.40	2.25	1.74	6.71
2017	14.36	50.47	0.75	2.49	1	1.69	0.00
2018	10.83	69.02	4.75	8.79	4.25	2.08	14.96
2020	12.47	70.77	5.5	4.34	3	1.44	21.11
r ^2^	−0.60	0.91	0.95	0.66	0.86	−0.015	
*p*-value ^2^	0.20	0.01	0.003	0.15	0.037	0.97	

^1^ Climatic values during the critical period (March and April): temperature (T), relative humidity (RH), rainfall (R), mean number of consecutive days in each fortnight of March and April in which the percentage of RH was above 80% (MCD-RH ≥ 80), mean number of days in each March and April fortnight with rain fall ≥10 mm (MD-R ≥10), and wind speed ^2^ Correlation analysis: r = correlation coefficient and *p*-value.

## Data Availability

The data are available in the Appendix A.

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
