# Peer review of "Epidemiological Studies of Brown Rot in Spanish Cherry Orchards in the Jerte Valley"

_jof, 2021, doi:10.3390/jof7030203_

Round 1

Reviewer 1 Report

Page 1 row 27: Monilinia laxa (Aderh. & Ruhland) Honey and Monilinia fructigena (Pers.) Honey

Page 1 row 29: Prunus avium L.

Page 3 row 114, 115: "on which Monilinia conidiophores and conidia were seen and/or sporulated" What do you mean with this? If fungus produces conidia it means it sporulates so you can delete "and/or sporulated"

Page 5 row 186: "M. laxa" italic

Author Response

Reviewer: 1

Comment 1) Page 1 row 27: Monilinia laxa (Aderh. & Ruhland) Honey and Monilinia fructigena (Pers.) Honey

Answer to comment 1) Corrected.

Comment 2) Page 1 row 29: Prunus avium L.

Answer to comment 2) Corrected.

Comment 3) Page 3 row 114, 115: "on which Monilinia conidiophores and conidia were seen and/or sporulated" What do you mean with this? If fungus produces conidia it means it sporulates so you can delete "and/or sporulated"

Answer to comment 3) Corrected.

Comment 4) Page 5 row 186: "M. laxa" italic

Answer to comment 4) Corrected.

Reviewer 2 Report

This study is aimed to investigate the relationship between cherry brown rot and weather conditions in the Jerte Valley in a long term (6 years) study. The study is designed in an approriate way and contains valuable and new scientific information on brown rot epidemiology. However, there are some points that need to be revised. The study can be suitable for publication after suitable revisions.

Suggestions:

Abstract: Please provide specific results in the Abstract (the real results provided only in L20-23). Please also give one sentence conclusion at the end of the Abstract.

L58: A space is missing between ’[12].’ and ’Since’.

L99: Please give a citation here or give information on organic disease and pest control.

L120 and L128: L-1 – fomatting.

L132: Please give more specific info on browen rot evaluation: e.g. assessment dates, sample sizes, replications.

L175: Y axis: do you mean ’Monilinia spp. incidence (%)’? Line is missing for Y axis.

L219: What are the bars represent? Give indication in the title of Figure 3.

L186: M. laxa – in italic

L193: experimental

L216: Y axis: do you mean ’brown rot incidence (%)’? Line is missing for Y axis.

L219: What are the bars represent? Give indication in the title of Figure 3.

L264: Delete double points ’(Table 5)..’

L266: M. laxa. and not Monilinia laxa.

L280: Monilinia spp.

L357: Give a conclusion section with 3-4 main conclusion points.

L380: Szoke, S.; Abonyi, F.

Author Response

Reviewer: 2

Comment 1) Abstract: Please provide specific results in the Abstract (the real results provided only in L20-23).

Please also give one sentence conclusion at the end of the Abstract.

Answer to comment 1) The abstract has been modified to include specific results and more conclusions of the work.

Comment 2) L58: A space is missing between ’[12].’ and ’Since’.

Answer to comment 2) Corrected.

Comment 3) L99: Please give a citation here or give information on organic disease and pest control.

Answer to comment 3) We have included information about organic disease and pest control

Comment 4) L120 and L128: L-1 – fomatting.

Answer to comment 4) Corrected.

Comment 5) L132: Please give more specific info on browen rot evaluation: e.g. assessment dates, sample sizes, replications.

Answer to comment 5) This information has been included in the new version

Comment 6) L175: Y axis: do you mean ’Monilinia spp. Incidence (%)’? Line is missing for Y axis.

Answer to comment 6) Corrected graphic

Comment 7) L186: M. laxa – in italic

Answer to comment 7) Corrected.

Comment 8) L193: experimental

Answer to comment 8) Corrected.

Comment 9) L216: Y axis: do you mean ’brown rot incidence (%)’? Line is missing for Y axis.

Answer to comment 9) Corrected graphic.

Comment 10) L219: What are the bars represent? Give indication in the title of Figure 3.

Answer to comment 10) The bars indicate the standard error and this information has been included in the figure title.

Comment 11) L264: Delete double points ’(Table 5)..’

Answer to comment 11) Corrected.

Comment 12) L266: M. laxa. and not Monilinia laxa.

Answer comment 12) Corrected.

Comment 13) L280: Monilinia spp.

Answer to comment 13) Corrected.

Comment 14) L357: Give a conclusion section with 3-4 main conclusion points.

Answer to comment 14) Conclusion section has been included.

Comment 15) L380: Szoke, S.; Abonyi, F.

Answer to comment 15) Corrected.